

# Long-term patient observation after conservative treatment of carpal tunnel syndrome: a summary of two randomised controlled trials

Tomasz Wolny and  Pawel Linek

Institute of Physiotherapy and Health Sciences, Musculoskeletal Elastography and Ultrasonography Laboratory, The Jerzy Kukuczka Academy of Physical Education, Katowice, Poland

## ABSTRACT

**Background**. Physiotherapy of carpal tunnel syndrome (CTS) involves manual therapy based on neurodynamic techniques. Until now, two randomized controlled trials have shown that immediately after therapy, CTS patients who received neurodynamic techniques had significant improvement in nerve conduction, pain, symptom severity (SSS), functional state (FSS), muscle strength (MS) and two-point discrimination (2PD). However, long-term effects seem to be more important, as they are the only ones that can significantly improve the patient's health and influence economic and social costs. Thus, the objective of this study was to evaluate the long-term (six months) effects of neurodynamic techniques in the conservative treatment of CTS patients.

**Methods**. Carpal tunnel syndrome patients (107) from two previously published randomised clinical trials were observed for six months after the treatment based on neurodynamic techniques.

**Results**. The sensory conduction velocity, motor conduction velocity, and motor latency were not subject to statistically significant changes within six months after therapy ($p > 0.05$). In both groups, there was further pain reduction ($p < 0.05$). In Group B, the symptom severity improved significantly ($p < 0.05$), while the functional status in both groups remained unchanged ($p > 0.05$). In both groups, there was muscle strength improvement ($p < 0.05$). Two-point discrimination remained unchanged six months after the therapy.

**Conclusion**. The use of manual therapy based on neurodynamic techniques maintains the beneficial effects 6 months after therapy in CTS patients.

Corresponding author
Pawel Linek, p.linek@awf.katowice.pl, Linek.fizjoterapia@vp.pl

## INTRODUCTION

Carpal tunnel syndrome (CTS) is the most common peripheral neuropathy (*Aroori & Spence, 2008*; *Biernawska, Niemczyk & Pierzchala, 2005*). In the initial period of the disease, there are temporary subjective symptoms (i.e., pain, night-time paraesthesia, numbness, and tingling in the area of the medial nerve innervation), which escalate

with time. Then, the disorders of various types of sensation develop, and in acute forms, there is impairment of the motor function of the hand (*Wolny et al., 2015*; *Wolny et al., 2016*; *Chang et al., 2008*). This leads to a reduction in the mobility of the hand, which can adversely affect working life, daily activities and overall health (*Wolny et al., 2016*; *Chang et al., 2008*; *Wolny, Linek & Michalski, 2017*; *Wolny, Saulicz & Linek, 2017*). *Thiese et al. (2014)* report that the incidence of CTS ranges from 6.3% to 11.7% and the occurrence of CTS is considered to be epidemic (*Thorne, 2007*). CTS is also common in the working age (*Newington, Harris & Walker-Bone, 2015*), which is directly connected with significant health-care costs and economic burden that are associated with lower productivity. The median of lost work time associated with CTS is 27 days per year, i.e., longer than any other (except fractures) work related disorder (*Foley, Silverstein & Polissar, 2007*). Therefore, CTS is the most expensive musculoskeletal disorder of the upper limb, whose estimated cost of medical care in the United States exceeds $ 2 billion a year—mainly due to surgical procedures (*Stapleton, 2006*). Thus, high economic (including surgical treatment) and social costs connected with CTS require exploration for cost-effectiveness and effective (long-term) treatments for this neuropathy.

The authors of scientific reports, as well as clinicians (including the American Academy of Neurology and the American Academy of Orthopaedic Surgeons), emphasise that conservative treatment should be used as first-line therapy, and only after the ineffectiveness of such treatment should surgical treatment be considered (*Gerritsen et al., 2002*). Frequently, conservative CTS treatment is based on physiotherapeutic management, in which procedures such as laser, ultrasound and manual therapy (including neurodynamic techniques) are applied (*Giele, 2001*). Recently, two randomised controlled trials have shown that immediately after 10 weeks of treatment (20 sessions), patients who received manual therapy based on neurodynamic techniques had significant improvement immediately after therapy (*Wolny & Linek, 2019*; *Wolny & Linek, 2018*). Wolny and Linek (*Wolny & Linek, 2019*; *Wolny & Linek, 2018*) showed that neurodynamic techniques positively influenced outcomes, such as nerve conduction, pain, subjective symptoms (SSS) and functional state (FSS) (Boston Carpal Tunnel Questionnaire—BCTQ), and the perception of two-point discrimination (2PD). To date, *Rozmaryn et al. (1998)* and *Bardak et al. (2009)* revealed that after the application of the therapeutic program, which consisted of manual therapy based on neurodynamic techniques performed as an autotherapy program, a significant improvement in the clinical condition in the long-term assessment of CTS patients was maintained. However, it should be noted that in studies conducted by *Rozmaryn et al. (1998)* the assessment was retrospective and concerned only the subjective improvement of clinical condition and the decrease in the number of surgical procedures performed. In the study by *Bardak et al. (2009)* the group in which only neurodynamic techniques were used achieved the poorest therapeutic effect. *Akalin et al. (2002)* comparing groups of CTS patients subject to the study showed that after the application of orthosis alone (group 1) and orthosis in combination with manual therapy based on neurodynamic techniques (performed as an autotherapy program) (group 2), the difference between the groups studied (in favour of group 2) only concerned the pinch grip after eight weeks. Thus, there are some conflicting opinions regarding

the long-term effects of the therapy after the application of manual therapy based on neurodynamic techniques in CTS patients. This may be due to a number of factors, such as the use of different therapeutic programs in which neurodynamic techniques were only one of the elements of a comprehensive therapy, the use of neurodynamic techniques as autotherapy, different duration of the therapy, the use of different research tools and the evaluation of other parameters of the clinical status of CTS patients.

Manual therapy based on neurodynamic techniques aims to improve nerve slide in the carpal tunnel, which is disturbed in CTS patients (*Greening et al., 2001*; *Erel et al., 2003*; *Shacklock, 2005*). It is even written that CTS is an "entrapment syndrome", which suggests a problem of the nerve with its free slide in relation to the surrounding tissues. Restoration of the dynamic balance between the relative motion of the nerve and the surrounding tissues may decrease the pathological forces, mainly compressive, which in turn will improve the physiological function of the nerve (*Fernández-de Las-Peñas et al., 2017*). This improvement in neuromechanics after the application of neurodynamic techniques may therefore further improve the physiology of the nerve, even after the conclusion of the therapy. However, there is a possibility that the effects of the therapy will persist only at the moment of continuous mechanical stimulation of the nerve, and shortly after the therapy cycle (no further stimulation of the nerve), the nerve slide becomes impaired again and CTS symptoms will reappear. Thus, to fully illustrate the effect of neurodynamic techniques, it seems reasonable to assess the patient's condition not only immediately after the therapy cycle but also during a longer observation period.

Hypothetically, it can be assumed that the therapeutic effect obtained directly after a 10-week cycle of CTS therapy according to two recently published studies (*Wolny & Linek, 2019*; *Wolny & Linek, 2018*) (a) will be perpetual, (b) will continue to improve as a result of restoring the proper mechanics and physiology of the median nerve (and thus its possible regeneration) and (c) will deteriorate as a result of the lack of continuous mechanical stimulation of the nerve. From a therapeutic point of view, long-term effects seem to be more important (compared to the effects immediately after the end of the therapy cycle), as they are the only ones that can significantly improve the health of patients and really influence economic and social costs. Thus, further patients' observation after treatment procedures seems reasonable and warranted. *Fernández-de Las-Peñas et al. (2017)* demonstrated that manual therapy and surgery had similar effectiveness for improving self-reported function, symptom severity, and pinch-tip grip force in CTS patients, but manual therapy has been found to be less costly than surgery in CTS treatment. Therefore, the objective of this study was to evaluate the outcomes changes six months after manual therapy based only on the neurodynamic techniques treatment of CTS patients. In some studies, the follow-up 6 months period is considered as a long-term (*Dakowicz et al., 2011*; *Fernández-de Las-Peñas et al., 2019a*; *Fernández-de Las-Peñas et al., 2019b*).

## MATERIALS & METHODS

### Ethics

The study was authorized by the Bioethics Committee for Scientific Studies at the Academy of Physical Education in Katowice on 08 March 2012 (Decision No. 7/2012), the annex on 28 February 2017 (No. KB/6/17). All study procedures were performed according to the Helsinki Declaration of Human Rights of 1975, modified in 1983. The clinical trial was registered at the Australian New Zealand Clinical Trials Registry (ANZCTR), number ACTRN12617000672358. All patients were informed about what the study would involve and told that they could withdraw at any stage without giving a reason. This study was conducted without a control group, as it is unethical to leave CTS patients without treatment for six months. Written informed consent was obtained from all participants.

### Study design

This study was an observational study in which patients from two previously published randomised clinical trials (*Wolny & Linek, 2019*; *Wolny & Linek, 2018*) were observed for six months after their last treatment procedure. In both studies, manual therapy based on neurodynamic techniques was compared to no treatment (*Wolny & Linek, 2019*) and 'sham' therapy (*Wolny & Linek, 2018*) in CTS patients, and the outcomes (nerve conduction, pain, symptom severity, functional status, 2PD, and muscle strength (MS) were collected and analysed immediately after the last treatment. Patients who received manual therapy based on neurodynamic techniques were then observed for six months after the conclusion of therapy, when all mentioned outcomes were measured again. In the first (*Wolny & Linek, 2019*) and second (*Wolny & Linek, 2018*) study, the sample size was calculated based on results from 10 and 20 participants, respectively. In both cases, the calculation of sample size was based on an alpha of 0.05 and a statistical power of 0.8. The study reported here was prepared in accordance with CONSORT guidelines (when applicable).

### Participants

The long-term observation are presented separately for CTS patients from the first study (Group A (*Wolny & Linek, 2019*)) and CTS patients from the second study (Group B (*Wolny & Linek, 2018*)) in such a way that the results can undergo intra-group and inter-group comparisons. Immediately after the therapy 58 CTS patients in Group A and 78 CTS patients in Group B were evaluated. A detailed scheme of the participants' flow is presented in Fig. 1. Basic characteristics and biometric features of patients in both groups are presented in Table 1.

### Diagnostic criteria for CTS

The CTS diagnosis was made by a physician on the basis of data collected from the interview, nerve conduction study (NCS), and clinical examinations. Based on the NCS, only participants who had diminished nerve conduction values ( $< 50$ m/s) and increased motor latency ($> 4t(ms)$) were included. The clinical diagnosis of CTS was based on the criteria that *Chang et al. (2008)* proposed:

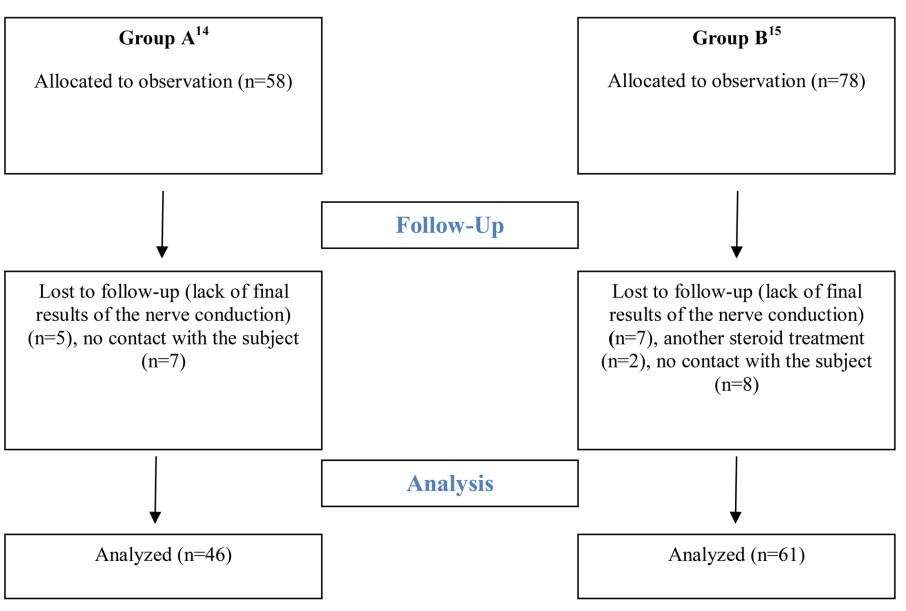

Figure 1  Flow diagram.

Table 1  Basic characteristics of the examined population.

| Characteristics | Group A (n = 46) | | | Group B (n = 61) | | |
|---|---|---|---|---|---|---|
| | Baseline[a] | 6 months later | P-value[b] | Baseline[a] | 6 months later | P-value[b] |
| Women (%) | 40 (87) | 40 (87) | NA | 40 (87) | 40 (87) | NA |
| Age (SD) year | 54.6 (8.4) | 54.7 (8.3) | 0.93 | 54.1 (9.3) | 54.2 (9.3) | 0.93 |
| Body mass (SD) kg | 70.4 (11.1) | 70.5 (10.8) | 0.96 | 70.4 (11.1) | 70.5 (10.8) | 0.96 |
| Height (SD) cm | 163.4 (6.88) | 163.6 (6.86) | 0.89 | 163.5 (6.5) | 163.4 (6.6) | 0.98 |
| BMI (SD) kg/m$^2$ | 26.6 (4.17) | 26.7 (4.12) | 0.90 | 26.6 (4.71) | 26.8 (4.57) | 0.92 |
| CTS symptoms: | | | | | | |
| —Unilateral (%) | 34 (74) | 34 (74) | NA | 45 (74) | 45 (74) | NA |
| —Bilateral (%) | 12 (26) | 12 (26) | NA | 16 (26) | 16 (26) | NA |

Notes.
[a] Data immediately after 10 weeks of treatment from published articles (Group A —*Wolny & Linek (2019)*; Group B —*Wolny & Linek (2018)*).
[b] 't'-test for dependent variables.
n, number of participants; NA, not applicable; CTS, carpal tunnel syndrome; BMI, body mass index.

Numbness and tingling in the area of the median nerve;

1. Night-timeparaesthesia;
2. PositivePhalen's test;
3. PositiveTinel'ssign;
4. Pain in the wrist area radiating to the shoulder

According to these criteria, the diagnosis of CTS was based on the presence of two or more symptoms (*Chang et al., 2008*).

The exclusion criteria were lack of consent, lack of cooperation from the patient, previous conservative, surgical or pharmacological therapy, cervical radiculopathy,

diabetes, rheumatoid disease, pregnancy, past trauma to the wrist, and muscular atrophy of the thenar eminence.

## Outcome measures

The NCS was performed in the same laboratory where the tests were performed immediately after the therapy (*Wolny & Linek, 2019*; *Wolny & Linek, 2018*). All test procedures were consistent with previous studies (*Wolny & Linek, 2019*; *Wolny & Linek, 2018*). Neuro-Mep electrodiagnostic equipment was used to perform the examinations, using an antidromic method with superficial electrodes. The temperature in the room where the test was performed was 24–26 °C. Before examination, patients were acclimated for 10 to 15 min. The skin temperature was measured by means of a surface thermometer and fluctuated between 32–34 °C. The assessment of nerve conduction results was based on the same normative values recommended by the laboratory where the following tests were performed: sensory conduction velocity ≥50 m/s, motor conduction velocity ≥50 m/s, and distal motor latency ≤4.0 t(m/s).

The pain was measured with the Numerical Pain Rating Scale (NPRS) (where "0" = no pain and "10" = maximum pain) (*Jensen et al., 1999*). Patients were asked to report the greatest pain in the CTS limb that they had felt during the last six months.

Assessment of the intensity of symptoms and disorders of hand functions was made with the BCTQ (*Levine et al., 1993*). This questionnaire assessed the subjective symptoms reported by the patient (SSS) and the functional state (FSS) of the patient. In patients with bilateral CTS, the questionnaire was performed separately for each hand.

MS in both the cylindrical and pinch grip tests was evaluated using the Jamara dynamometers according to the recommendations of the American Society of Hand Therapists (*Watanabe et al., 2005*). During the test, the dynamometer was placed between the metacarpus and through the fifth fingers (cylindrical grip) and between the thumb and the lateral surface of the index finger (pinch grip). The values were in kilograms of strength (kg). The measurement was done three times, and the mean value was added to the analysis (*Wolny & Linek, 2019*; *Wolny & Linek, 2018*).

The examination of 2PD was performed using a standardized Dellon discriminator (*Crosby & Dellon, 1989*). The 2PD study used the procedure proposed by *Wolny, Linek & Michalski (2017)*. The measurement was done three times, and the mean value was added to the analysis.

## Intervention

Physiotherapy for both groups was based on neurodynamic techniques directed at the median nerve. Both sliding and tensioning techniques were used. The following techniques were used: one-direction proximal sliding mobilisation, one-direction distal sliding mobilisation, one-direction proximal tensioning mobilisation and one-direction distal tensioning mobilisation. The standard protocol consisted of three series of 60 repetitions of sliding and tensioning neurodynamic techniques separated by inter-series intervals of 15 s, twice a week for 20 sessions. A detailed description of the techniques used (including photos) can be found elsewhere (*Wolny & Linek, 2019*; *Wolny & Linek, 2018*).

## Statistical analysis

Analysis of the collected data was performed using the Statistica 13.1 software package. Data collected immediately after 10 weeks of treatment was compared with data collected after six months of follow-up by dependent (comparisons inside groups A and B) and independent $t$-tests (comparisons between groups A and B). For nominal data, the Chi-Square ($\chi^2$) test was used. The critical p-level was set to 0.05.

## RESULTS

Sensory conduction velocity, motor conduction velocity and motor latency in both groups A and B were not subject to statistically significant changes six months after the conclusion of therapy. In Group A differences in sensory conduction velocity, motor conduction velocity and motor latency was 1.7 (95% CI [2.19–5.73]), 2.1 (95% CI [1.11–5.25]) and 0.08 (95% CI [0.15–0.31]) respectively. In Group B, differences in sensory conduction velocity, motor conduction velocity and motor latency was 2.1 (95% CI [1.25–5.27]), 1.3 (95% CI [0.49–3.13]) and 0.07 (95% CI [0.14–0.27]) respectively. In both analysed groups, there were significant differences in pain between the results obtained immediately after the end of the therapy and after six months of observation. In groups A and B, pain reduction was 0.34 (95% CI [0.09–0.39]) and 0.34 (95% CI [0.18–0.51]), respectively. In the assessment of subjective symptoms, the comparison of results obtained immediately after the end of therapy and six months later in Group A did not show statistically significant differences compared to Group B. These differences occurred only in Group B where pain reduction was 0.23 (95% CI [0.13–0.32]). In the assessment of function, the comparison of results obtained immediately after the end of therapy and six months later in both groups did not show statistically significant differences. MS in both examined grips (i.e., cylindrical and pinch) improved in both examined groups six months after the conclusion of the therapy. In Group A, the strength of the pinch and cylindrical grip tests improved by 0.53 kg (95% CI [0.26–0.81]) and 1.1 kg (95% CI [0.39–1.84]), respectively. In Group B, the average improvement was 0.43 kg (95% CI [0.13–0.84]) for the pinch grip and 0.8 kg (95% CI [0.24–1.61]) for the cylindrical grip. The 2PD sensation did not show statistically significant changes in Group B, comparing the results immediately after the conclusion of the therapy and 6 months later. In Group A, the discriminatory sensation was not studied. Detailed data are presented in Table 2. Additionally, the comparative analysis carried out between groups A and B did not show statistically significant differences (in all cases $P > 0.05$).

## DISCUSSION

The primary objective of this study was to evaluate the long-term effects of manual therapy based on the application of neurodynamic techniques in CTS patients. The present study identified that both groups were significant differences in pain and MS at follow-up 6 months. The assessment of subjective symptoms of Group B showed significantly different at follow-up 6 months. However, comparable outcomes of sensory conduction velocity, motor conduction velocity, motor latency and assessment of cunduction were

**Table 2 Outcomes measured in the examined population.**

| Characteristics | Group A (*n* = 46) | | | Group B (*n* = 61) | | |
|---|---|---|---|---|---|---|
| | Baseline[a] | 6 months later | *P*-value[b], 95% CI | Baseline[a] | 6 months later | *P*-value[b], 95% CI |
| SCV (SD) m/s | 37.8 (10.9) | 36.1 (10.3) | 0.37 (2.19–5.73) | 36.5 (10.3) | 34.4 (10.7) | 0.22 (1.25–5.27) |
| MCV (SD m/s | 55.9 (7.21) | 53.8 (9.75) | 0.21 (1.11–5.25) | 55.7 (6.36) | 54.4 (5.29) | 0.15 (0.49–3.13) |
| MT (SD) t(ms) | 4.53 (0.67) | 4.45 (0.57) | 0.53 (0.15–0.31) | 4.46 (0.72) | 4.39 (0.63) | 0.52 (0.14–0.27) |
| NPRS (SD) (0–10) | 1.38 (1.04) | 1.04 (0.57) | 0.00[*] (0.09–0.39) | 1.41 (1.04) | 1.07 (0.77) | 0.00[*] (0.18–0.51) |
| BCTQ-SSS (SD) | 1.81 (0.46) | 1.73 (0.45) | 0.31 (0.08–0.25) | 1.71 (0.45) | 1.48 (0.35) | 0.00[*] (0.13–0.32) |
| BCTQ-FSS (SD) | 2.02 (0.68) | 1.81 (0.55) | 0.06 (0.01–0.44) | 1.95 (0.61) | 1.82 (0.49) | 0.15 (0.04–0.31) |
| MS CG (SD) kg | 28.4 (6.32) | 29.5 (5.89) | 0.00[*] (0.39–1.84) | 27.3 (6.22) | 28.1 (5.33) | 0.00[*] (0.24–1.61) |
| MS PG (SD) kg | 7.92 (1.46) | 8.45 (1.61) | 0.00[*] (0.26–0.81) | 7.88 (1.55) | 8.31 (1.34) | 0.00[*] (0.13–0.84) |
| 2PD FI (SD) | NA | | | 4.04 (0.86) | 3.81 (0.69) | 0.14 (0.01–0.47) |
| 2PD FII (SD) | | | | 3.43 (0.82) | 3.21 (0.75) | 0.07 (0.02–0.46) |
| 2PD FIII (SD) | | | | 3.38 (0.88) | 3.45 (0.91) | 0.84 (0.84–0.69) |

**Notes.**
[a] Data immediately after 10 weeks of treatment from published articles (Group A —*Wolny & Linek (2019)*; Group B—*Wolny & Linek (2018)*).
[b] '*t*'- test for dependent variables.
[*] statistically significant difference.

SCV, sensory conduction velocity; MCV, motor conduction velocity; MT, motor latency; NPRS, Numerical Pain Rating Scale; BCTQ, Boston Carpal Tunnel Questionnaire; SSS, Symptom Severity Scale; FSS, Functional Status Scale; MS CG, Muscle Strength Cylindrical Grip; MS PG, Muscle Strength Pincer Grip; 2PD, Two–Point Discrimination Sense; FI, FII, FIII, Finger I, II, III; NA, not applicable.

disclosed between both groups. Therefore, it can be concluded that manual therapy based on neurodynamic techniques has beneficial effects in the physiotherapy of patients with mild and moderate CTS not only in the evaluation immediately after therapy but also in the long-term evaluation (six months).

The maintenance of different parameters of nerve conduction, function and 2PD sense at the same level indicates a beneficial therapeutic effect. However, those parameters (sensory conduction velocity, motor conduction velocity, distal motor latency, functional status, and 2PD sense) did not reach normative values after 20 therapeutic sessions, which may suggest that: (a) the therapy (nerve stimulation) did not last long enough and (b) the therapy applied reduced, but did not (completely) remove, the adverse changes occurring in the nerve. The six—month observation showed that patients undergoing the given therapy experienced further significant pain reduction, which may result from the improvement of nerve mechanics immediately after the therapy. The pain before the beginning of the therapy lasted for many months, so it could already have its central representation, as suggested by some authors (*Barr, Barbe & Clark, 2004*). Therefore, a longer observation period was necessary for the change (reduction) of pain. A significant increase in MS (despite the lack of further therapy) may be related to the pain reduction achieved. *Tamburin et al. (2008)* demonstrated that pain is a factor contributing to weakening MS. Reduction of pain probably resulted in more frequent use of the hand (during various daily activities), which naturally contributed to an increase in the strength of the muscles releasing both the cylindrical and pinch grip tests.

A number of authors evaluated the long-term effects of manual therapy based on the application of neurodynamic techniques in CTS patients. *Rozmaryn et al. (1998)* in

a clinical and control retrospective study in which 197 CTS patients (divided into two groups) were examined, evaluated the therapeutic program that included orthosis, non-steroidal anti-inflammatory drugs and steroid injections, as well as variable temperature baths. In the experimental group, CTS patients were additionally instructed on how to perform neurodynamic techniques on their own, as an autotherapy program. After an average period of 23 months, 70.2% of the subjects declared positive effects of the therapy, and only 19.2% reported that the symptoms of CTS remained unchanged. In their conclusions, the authors emphasised that the use of the therapeutic program presented above may significantly reduce the number of patients subject to surgical treatment. On the other hand, *Bardak et al. (2009)* evaluated 11 CTS patients, who were divided into three groups (group 1—therapy was based on steroid injections and orthosis; group 2—therapy was based on steroid injections, orthosis and neurodynamic techniques as an autotherapy program; and group 3—only neurodynamic techniques as an autotherapy program), and therapeutic effects were evaluated 11 months after the conclusion of therapy. Significant functional improvement and symptom reduction were achieved in all groups, although the results in the group where only neurodynamic techniques were used, were significantly weaker. In another study, *Akalin et al. (2002)* evaluated 28 CTS patients, who were divided into two groups. In both groups, the therapy consisted of putting on an orthosis overnight and wearing it as long as possible during the day (for four weeks). Additionally, in the experimental group, patients were instructed on how to perform neurodynamic techniques on their own, as an autotherapy program. In the final evaluation, after eight weeks of therapy, a statistically significant improvement of all parameters was achieved in both groups. In the experimental group (in which neurodynamic techniques were used), a significant difference from the group wearing just the orthosis occurred only in the assessment of the pinch grip in favour of the experimental group.

Contrary to the present study, the results of the studies presented in the above paragraph do not clearly indicate beneficial long-term therapeutic effects of the application of neurodynamic techniques in CTS patients. This is probably due to the different methods of therapeutic management since neurodynamic techniques were only an additional element of the therapeutic program and were always used as an autotherapy program, which makes it difficult (impossible) to constantly control the regularity and correctness of the mobilisations performed. This study (as well as recently published work by *Wolny & Linek (2019)* and *Wolny & Linek (2018)*, shows that neurodynamic techniques performed by a qualified physiotherapist have a positive therapeutic effect not only immediately after the therapy but also during the six-month observation period. In assessing the effectiveness of any manual therapy (including neurodynamic techniques), the manner in which the therapy is conducted is crucial, as only a trained person can guarantee the appropriate technique, control and regularity of the conducted therapy. In every autotherapy (including neurodynamic techniques), it is difficult to control the variables that are unquestionable in the methodology of scientific research. Thus, the assessment of the effectiveness of neurodynamic techniques used in the form of autotherapy is questionable.

So far, the literature has lacked studies evaluating the long-term effects of manual therapy based on neurodynamic techniques, which would be the only form of therapy in CTS patients. Therefore, the conducted studies are the first to present beneficial effects of manual therapy based on neurodynamic techniques in the evaluation of long-term effects of physiotherapy on CTS patients. The high value of such studies is the evaluation of both objective indicators, such as nerve conduction, 2PD sense, MS and subjective sensations of CTS in patients related to pain reduction, symptom severity reduction and improvement of function. The results of the research allow us to conclude that manual therapy based on neurodynamic techniques has beneficial therapeutic effects in the conservative treatment of mild and moderate forms of CTS both in the short-term evaluation (*Wolny & Linek, 2019*; *Wolny & Linek, 2018*) and in the evaluation of long-term therapeutic effects.

## LIMITATIONS

The lack of a control group is a major flaw of the present study because important other variables like spontaneous recovery, Hawthorne effects or placebo effects would contribute to the outcomes observed. Thus, the lack of a control group means that we cannot estimate true therapeutic effects in isolation from other effects. For ethical reasons, we were unable to leave patients without any therapy for such a long period of time. However, we believe that after successful treatment, the patients should be further observed on how their outcomes change in the long term. From a therapeutic point of view, distant therapeutic results seem to be more important (compared to the effects immediately after the end of the therapy cycle), as they are the only ones that can significantly improve the general health of patients and really influence economic and social costs.

## CONCLUSIONS

Six months after the last session of manual therapy based on neurodynamic techniques parameters of nerve conduction(sensory conduction velocity, motor conduction velocity and distal motor latency) and functional state were at a similar level to the results gained immediately after therapy. Within six months after the application of neurodynamic techniques, further pain reduction and improvement in the strength of the muscles engaged during both the pinch and cylindrical grip tests were observed in CTS patients.

### Funding
The authors received no funding for this work.

### Competing Interests
The authors declare there are no competing interests.

## Author Contributions

- Tomasz Wolny conceived and designed the experiments, performed the experiments, analyzed the data, contributed reagents/materials/analysis tools, prepared figures and/or tables, authored or reviewed drafts of the paper, approved the final draft.
- Pawel Linek conceived and designed the experiments, analyzed the data, contributed reagents/materials/analysis tools, authored or reviewed drafts of the paper, approved the final draft.

## Human Ethics

The following information was supplied relating to ethical approvals (i.e., approving body and any reference numbers):

The study was authorized by the Bioethics Committee for Scientific Studies at the Academy of Physical Education in Katowice on 08 March 2012 (Decision No. 7/2012), the annex on 28 February 2017 (No. KB/6/17).

## Data Deposition

The raw measurements are available as a Supplemental File.

## Supplemental Information

Supplemental information for this article can be found online at http://dx.doi.org/10. 7717/peerj.8012#supplemental-information.

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
