# Peer review of "Long-term patient observation after conservative treatment of carpal tunnel syndrome: a summary of two randomised controlled trials"

_PeerJ, doi:10.7717/peerj.8012_

## Round 0.1 · original submission · Major Revisions

Dear authors,

Thank you very much for submitting your work to PeerJ. There are several issues to be address before this work is suitable to publish.

Please read carefully the comments from the reviewers, make revisions accordingly, and additionally provide a point-by-point response to reviewers' comments.

With kind regards,

Liang Gao, MD, PhD
Academic Editor, PeerJ

Reviewer 1 ·

Basic reporting

no comment

Experimental design

no comment

Validity of the findings

no comment

Additional comments

#39930
Long-term patients observation after conservative treatment of carpal tunnel syndrome: a summary of two randomised controlled trials

Overall, this is a manuscript for evaluating the outcomes changes six months after manual therapy based only on the neurodynamic techniques treatment of CTS patients. The authors performed an observational study and concluded that the use of manual therapy based on neurodynamic techniques maintains the beneficial effects 6 months after therapy in CTS patients.

The topic is quite interesting and clinical important that can significantly improve the patients’ health and influence economic and social costs. However, this manuscript is not well written and several issues need to be addressed before the article is suitable for publication.

The main issues of this manuscript includes:

1. In the Results, your most important issue is that the table didn’t include (95% CI). Another issue is that real and accurate data (numbers and p values) should be reported.

2. Follow-up 6 months means long-term? Please find some references support your opinion.

3. The discussion is too prolix. I suggest that your write paragraph 1 like this:

The present study identified that both groups were significant differences in pain and MS at follow-up 6 months. The assessment of subjective symptoms of group B showed significantly different at follow-up 6 months. However, comparable outcomes of sensory conduction velocity, motor conduction velocity, motor latency and assessment of cunction were disclosed between both groups.

The detail issues are as follows:

Line 147: This work... This study...

Line 195: Please provide detailed laboratory testing gear.

Line 234-236, 239-241: Please write real and accurate data (numbers and p values) even the results not significant difference.

Line 238: ... pain reduction was 0.34 (95% CI, 0.09–0.39) and 0.34 (95% CI, 0.18–0.51) ... ... pain reduction was 0.34 (95% CI, 0.09 - 0.39) and 0.34 (95% CI, 0.18 - 0.51) ... Please check full-text and modify them.

Line 246: ... the strength of the pinch and cylindrical grip tests improved by 0.53 kg (95% CI, 0.26–0.81) and 1.1 kg (95% CI, 0.39–1.84), respectively. ... ...the strength of the cylindrical and pinch grip tests improved by 1.1 kg (95% CI, 0.39–1.84) and 0.53 kg (95% CI, 0.26–0.81), respectively...

Lines 253: ... (in all cases p>0.05) ... ... (in all cases P > 0.05) ... Please check full-text and modify them.

Lines 380-384: the style of references 1, 2, 3, 8 are different. Please modify them.

Table 1: ... Group A (n=46), Group B (n=61), 6months, p-value, Age(year), Body mass (kg), Height (cm), BMI (kg/m2) ... ... Group A (n = 46), Group B (n = 61), 6 months, P- value, Age (SD) years, Body mass (SD) kg, Height (SD) cm, BMI (SD) kg/m2...

Table 2: ... Group A , Group B, 6months, p-value, SCV(m/s), MCV(m/s), MT(t(ms)), NPRS (0-10), BCTQ-SSS, BCTQ-FSS, MS CG(kg), MS PG(kg) ... ... Group A (n = 46), Group B (n = 61), 6 months, P- value, SCV (SD) m/s, MCV (SD) m/s, MT (SD) t(ms), NPRS (SD) (0-10), BCTQ-SSS (SD), BCTQ-FSS (SD), MS CG (SD) kg, MS PG (SD) kg...

Reviewer 2 ·

Basic reporting

Literature references are old;
In the table, what does mean of the content in the bracket?

Experimental design

no comment

Validity of the findings

it's better to also compare the results in the sham group

Additional comments

The authors have extensively evaluate the long-term (six months) effects of neurodynamic techniques in the conservative treatment of CTS patients.The study could benefit from additional controls such as
the results change in the sham group.
Criticisms:
1) Literature references are old;
2) In the table, what does mean of the content in the bracket?
3) it's better to also compare the results change in the sham group.

---

## Round 0.2 · Minor Revisions

Dear authors,

Please make the several minor revisions noted before we can accept this work.

Best regards,

Liang Gao, MD, PhD
Academic Editor, PeerJ

Reviewer 1 ·

Basic reporting

no comment

Experimental design

no comment

Validity of the findings

no comment

Additional comments

Lines 181-182: ... values (<50 m/s) ...change to ... values (< 50 m/s) ... ; ...latency (>4t(ms)) ...change to ....latency (> 4t(ms)); Please check full-text and modify them.

---

## Round 0.3 · accepted · Accept

Dear authors,

Congratulations! I am very glad to notify you that your manuscript is accepted in the present version.

Best regards,

Liang Gao, MD, PhD
Academic Editor, PeerJ